# Romani Girls Matter: Developing a Participatory Action Research Protocol for Reproductive Justice

**DOI:** 10.3390/healthcare11050755

**Published:** 2023-03-04

**Authors:** Belen Soto-Ponce, Manuel Garcia-Ramirez, Lucía Jiménez

**Affiliations:** 1CESPYD, Department of Social Psychology, University of Seville, 41018 Sevilla, Spain; 2Department of Developmental and Educational Psychology, University of Seville, 41018 Sevilla, Spain

**Keywords:** Romani girls, reproductive justice, participatory action research, Photovoice, sense of mattering

## Abstract

Despite the last few decades’ advances towards social and gender justice, reproductive decisions are still a source of oppression for many European Romani women and girls. This protocol aims to propose a model to empower Romani women and girls’ reproductive decisions, inspired by Reproductive Justice—the recognition of women and girls’ ability to make safe and free decisions about their bodies and reproduction. Through Participatory Action Research, 15–20 Romani girls and their families, two Romani platforms, and key agents from a rural and an urban context in Spain will participate. They will (1) contextualize Romani women and girls’ inequities, (2) build partnerships, (3) implement Photovoice and advocate for their gender rights, and (4) assess the initiative’s related changes, using self-evaluation techniques. Qualitative and quantitative indicators will be collected to assess impacts among participants, while tailoring and assuring the quality of the actions. Expected outcomes include the creation and consolidation of new social networks, and the promotion of Romani women and girls’ leadership. For this, Romani organizations must be transformed into empowering settings for their communities, spaces where Romani women and girls assume responsibility of the initiatives, being these tailored to their real needs and interests, and guaranteeing transformative social changes.

## 1. Introduction

Health inequities suffered by European Romani women are regularly linked to their early reproductive decisions (RD) [1,2,3]. Known as the largest European ethnic minority group, it is estimated that 11 million Romani are living on the European continent. They are a young population; partly because they tend to have many children, and partly because they have a life expectancy 15 years lower than the rest of the population [4]. Even with the great advances that Europe has made in gender justice, today, many Romani women and girls (RWG)—just as centuries ago—are living in the margins of society [5]. RWG’s reproductive decisions have been clearly affected by antigypsyism policies, such as those that forced Romani women’s sterilizations in many European countries [6]. Moreover, other antigypsyism mechanisms convert RD into sources of structural oppression, including substandard housing conditions, unresponsive healthcare, violence and stigma, scarce education from early childhood, and vast obstacles to accessing a regular labor market. These mechanisms push some Romani girls to view their womanhood as being linked to marriage, rearing children, caregiving and housekeeping, and they end up viewing these as the only way to matter to their families and communities [7,8,9,10].

Huge efforts have been carried out in attempts to overcome gendered inequities. Often, these have been based on the assumption that solutions would come from RWG’s access to sex education, reproductive literacy, contraceptive measures, and family planning services [11]. Although the initiatives were based on scientific evidence and agreed upon by academics, policymakers, and service providers, they have not only been unable to reverse the issues, but they have sometimes aggravated them. In fact, instead of achieving the expected results, these initiatives have been rejected by Romani communities [11,12]. Strong evidence highlights that this rejection is due to the weak response to the urgent challenges that RWG confront daily; the lack of harmony with their cultural practices, traditions, and spirituality; and the disregard of Romani strengths developed to face adversity over centuries [13,14,15]. The lack of recognition of these issues as relevant dimensions to address Romani gendered inequities can be catalogued as slow violence [16]—understood as diverse and invisible acts entangled in societal, institutional, and community dynamics, such as discrimination, neglect, and marginalization of Romani communities—that perpetuates oppression and inequalities [17]. These processes lead to the development of new efforts mainly based on ethnocentric patterns, which end up ignoring the roots of RWG’s suffering, and consequently, lack of Romani communities’ support.

Such failures uncover the wicked nature of Romani reproduction inequalities, understood as inequities over which there is little agreement on their causes and the best way to address them, and that defy the capacity of any one organization to solve them [18,19]. The limited trust RWG have towards promoters of these initiatives (organizations, institutions, researchers) has provoked their withdrawal, even making them more refractory to future initiatives and feeding feelings of abandonment in the most isolated communities [14,20]. These examples show that Romani women—especially younger ones—were unable to progress in the understanding of the causes of their own suffering, perceiving themselves as entitled to define their life goals, as well as to find and weave alliances within their families and communities to advocate for their rights [21].

As in other health-based wicked inequities (e.g., obesity, tobacco use, COVID vaccination), to overcome reproductive inequities of RWG, there is need for innovative actions in which they become aware about the unfair conditions they, their families, and communities suffer; increasing intra- and intergenerational complicity and solidarity among women; and creating safe community settings for themselves. This article describes a Participatory Action Research protocol, inspired by Reproductive Justice, as the most suitable approach to build safe spaces that facilitate RWG’s empowerment towards their own reproductive decisions.

### 1.1. Reproductive Justice for Romani Women and Girls

Reproductive Justice (RJ) highlights that RWG’s wellbeing is connected to empowering them to freely make decisions in all their life domains including—but not limited to—their bodies, sexuality, and reproduction, which can be achieved only if they have the necessary resources, as well as sufficient economic, social, and political power [22,23]. On the one side, RJ posits that women and girls must be able to take informed decisions regarding their own lives and aspirations. On the other, it also implies that they must enjoy guaranteed access to resources that facilitate the pursuit of those decisions completely, freely, and safely, while allowing them to obtain recognition and influence [24]. Reproductive justice arises in the context in which the inequalities suffered by humanity and its environment have paved the way for new proposals driven by new actors claiming legitimacy to new rights. This is the case with structural racism [25], gender equity [26], environmental justice [27], displaced people’s justice [28], LGTBQ + rights [29], and indigenous justice [30], among others. All of the above highlight that current inequities are the result of the intersection of multiple oppressions at multiple levels which manifest as a unique expression in the suffering of silenced and emerging individuals and collectives in their local contexts and communities.

Based on the previous assumptions, an RJ approach for RWG highlights some key points. First, Romani girls should be provided with opportunities from the beginning of their lives to separate their mattering from the female traditional roles (i.e., marriage, childrearing, parenting, caregiving) while offering them alternatives and safe corridors to imagine and explore new and free aspirations. Second, it should provide tools and competencies to analyze the influence of power structures inside and outside their communities on their RD, and how these could be reproducing sociocultural structures that oppress them. Third, all RWG must have real opportunities to design their lives around their experiences and find and weave sufficient allies to accompany them, give guidance, protection, and emotional support. Fourth, they must have the tools to evaluate their experiences, make decisions, choose new allies for new targets, and have the strength to self-evaluate and know the impact of their actions on the environment.

### 1.2. Reproductive Justice and Participatory Action Research

Participatory Action Research (PAR) is especially relevant to accomplish the above strategies linked to the RJ approach. PAR should offer an ideal scenario to bring stakeholders interested in overcoming RWG inequities together, while the roles are redefined and distributed, power is balanced and shared among participants, and decisions are ultimately taken by RWG [21,31]. However, although PAR initiatives usually state that their purpose is to advance social justice, they often end up concentrating their efforts on providing assistance, capacity building, or ameliorative interventions [14]. A good example of this has been many initiatives designed and implemented to cope with the turbulences suffered by RWG communities during the COVID pandemic [32]. Far from offering resources to them to face new challenges and take advantage of new opportunities (e.g., digitalization of their younger ones’ activities), they have pushed them into households and have been unable to prevent them from taking on harsh housework. Online schooling has been an inaccessible pipe dream for many Romani girls; many will never return to school [33].

Some dimensions must be present for PAR to overcome the previously mentioned barriers and become RJ-inspired. RWG must lead the processes that shape the community context, the underlying determinants of inequities and the strengths they have to confront them. While RJ seeks the legitimization of RWG as political actors, PAR must provide opportunities for them to have time and space to recognize and perceive themselves as having the capacity to decide and influence [34,35]. PAR must ensure actions aimed at disentangling the complexity of communities so as not to repeat the same power asymmetries they combat. For example, it is not possible to overcome the conditions of inequality suffered by RWG without taking into account the impact that secularly established and sacralized traditions and customs has had on them [36].

In addition to the former challenge, RWG are often represented in PAR initiatives through community leaders and organizations. Are these representatives free of conflict of interests so that we can ensure that the actions they decide on address the needs of the women and not those of their organizations? RJ calls for ensuring that RWG have “presence, voice, and influence” [37] (p. 1973) in initiatives that affect their lives so that policy decisions are truly theirs. However, this is not a simple task. RWG suffer dramatic consequences of social inequalities and lack the skills to occupy effective positions in the political arena. The same mechanisms of oppression that co-opt their life projects prevent them from transcending their conditions of alienation, so that it is impossible for women who uncritically accept life patterns rooted in traditional gender roles to imagine alternative life projects. In contrast, a critical vision implies understanding the power dynamics that over time construct the world we know and in which we navigate in our daily lives [38,39]. RJ posits that PAR should aim to facilitate this journey toward constructing a critical view of everyday life and personal experiences [25,40,41,42].

Building a critical vision of their own realities demands that evaluation components must be placed at the heart of developed initiatives, so that RWG are impelled to (1) identify problems and goals that are important for them; (2) analyze, understand, and prioritize those that are more relevant; (3) choose and decide what actions need to be taken to achieve their proposed goals; and (4) take stock of the contribution of those actions towards their goals, while planning future steps to assure their real impact [43]. Through evaluation, RWG can use their voices to decide what is important to them, while also ensuring accountability of the actions implemented through the initiatives developed. Evaluation components can then become empowering methods for RWG to lead their own initiatives and, as a result, to lead their own life decisions. In summary, RJ impels PAR to build a new Romani womanhood assuming that they are new political actors who have new rights (e.g., civil, political, social, cultural) and need new scenarios for taking new actions (e.g., blogging, protesting, resisting, organizing) to inhabit new spaces in Romani communities [44,45].

Adopting a Reproductive Justice-based Participatory Action Research approach in promoting RWG reproductive decisions is consistent with a mature and sophisticated understanding of generating evidence when supporting children, young people, and families [46]. From this understanding, the development of evidence-based practices requires the best scientific evidence from applied science in combination with agreed professional experience. Moreover, what is critical in this approach is the reformulation of the “client with preferences” that are considered in the classical definition of evidence into “subjects of rights”. Recognizing intervention participants as rights owners involves establishing relationships between results and rights, a culturally sensitive approach, and placing the voices of children, young people, and families in the center [47].

### 1.3. Aim

The aim of this paper is to describe a Reproductive Justice Participatory Action Research protocol to empower Romani girls’ reproductive decisions, following the Standard Protocol Items: Recommendations for Interventional Trials (SPIRIT) guidelines [48].

This protocol is framed within a strategic framework to implement three initiatives that were funded at Spanish national and regional levels, to advocate for policies and social empowerment to overcome the disparities that RWG suffer. At the national level, call for proposals focused on developing solutions to the challenges of Spanish society; specifically, this initiative aimed to respond to challenges in the field of social psychology and feminist studies, which addressed the study of norms, prejudices, conflict, discrimination and social influence in various social contexts. At the regional level, call for proposals were also aimed at promoting knowledge oriented to the challenges of Andalusian society; in this case, the challenge of health and social well-being of marginalized groups. These call for proposals were framed within the objectives and challenges of the Horizon 2020, focusing on empowering vulnerable groups on the social determinants that limit their well-being. RWG are one of the central axes of many scientific initiatives since the persistent failure in improving their social inclusion in Spain and other European countries. Therefore, this protocol aims to respond to those priorities included within the funding strategies.

## 2. Materials and Methods

### 2.1. Study Design

This initiative follows PAR; specifically, participants will define their own goals and actions to be taken, in order to achieve those. In the first phase, they will assess problems, needs and assets of their communities, reflecting on their life conditions. They will then determine the actions to be implemented in order to advocate for their rights in their own terms (i.e., preparing exhibits to share on their communities). Lastly, they will evaluate the intervention and its results, taking part in data collection and its analyses [49,50,51]. Steps to be taken are described below.

The design follows a quasi-experimental evaluation multi-site approach with parallel groups (intervention and comparison groups) and two evaluation moments (pre-test and post-test), including process, implementation, and outcome indicators [52]. The comparison group will not participate in any of the proposed steps nor sessions. They will only participate in baseline and endline interviews, where qualitative and quantitative data will be collected.

### 2.2. Study Settings

This initiative will be carried out in different settings and facilities located around Romani local communities (e.g., civic centers, schools, streets, city council buildings, etc.) in the regions of Andalucía and Comunidad Foral de Navarra, in southern and northern Spain, respectively. In each region, partnerships and shared resources among local communities, civil society groups, local institutions, and research teams—including researchers from the University of Seville and the Public University of Navarra—will facilitate the proper implementation of the initiative.

### 2.3. Participants and Recruitment

At an individual level, 15–20 Romani girls and their families will be involved in the initiative in each of the local neighborhoods. For the intervention group, the inclusion criteria for Romani girls will be girls who: (1) are between 10 and 14 years old; (2) self-identify as Romani; (3) have no children; (4) are not pregnant; (5) are attending school; and (6) have untapped leadership potential and ability to work together. These data will be gathered through baseline interviews with Romani girls and their families, including questions regarding sociodemographic data, teamwork skills, and leadership potential. The exclusion criteria will be: (1) having comorbidities or health conditions that could bias the implementation or results of the intervention; (2) those who can commit to attending the sessions or being involved in the project only sporadically; and (3) those for which the intervention can increase risks of adverse events. The comparison group will be composed of Romani girls from similar contexts, with comparable family socio-economic statuses, and in the age range of the girls who will receive intervention. At organizational level, this initiative will include platforms that have strong influence within local Romani communities (Gaz Kalo and Yilo). They will oversee the transferring of their learned knowledge and previous experiences serving local Romani communities to facilitate the implementation of this study, as well as maximize the initiative’s impacts at community, institutional, and policy level. At a community level, contexts and neighborhoods entail diverse Spanish local contexts with high at-risk Romani populations, including marginalized rural contexts (i.e., Mancomunidad de San Adrián, in Comunidad Foral de Navarra) and marginalized urban contexts (i.e., the Torreblanca and Polígono Sur neighborhoods, in Seville).

### 2.4. Intervention: An Overview

RJ-based PAR will be co-designed, implemented, and evaluated along with Romani girls through group sessions within a psychosocial intervention facilitated by Romani women from their local communities. The intervention will be tailored by members of Romani grassroots organizations—specifically, GazKalo and Yilo, already working with participant Romani communities in Mancomunidad de San Adrián and Seville—and academic researchers from each context (researchers from Pamplona, Comunidad Foral de Navarra and researchers from Seville, Andalucía). To achieve this, self-evaluation strategies at organizational, community, and individual levels will be placed at the center of the four RJ-based PAR dimensions (see Table 1) in order for RWG to adopt leading roles in decision-making processes regarding the definition of their aims, actions to be taken, and expected outcomes.

#### 2.4.1. Contextualizing Romani Women and Girls’ Inequities

Contextualizing Romani girls’ inequities entails mapping Romani adolescents’ psychosocial dimensions linked to gendered identity roles (i.e., satisfaction with life, sense of mattering at family, school, and community levels, and socio-political agency), as well as their personal social support networks. At a community level, RWG’s life narratives will be mapped to identify strengths and the mechanisms of oppression that co-opt their RD. At an organizational level, programs, good practices, and materials developed to tackle RWG’s inequities will be mapped, including organizations’ influence. Narratives to identify empowering settings for its members and Romani communities will be collected.

#### 2.4.2. Building Partnerships, Alliances, and Solidarity

Adolescent girls will collaboratively consolidate existing alliances with their mothers (and families) and expand their community social ties and networks. Building partnerships, alliances, and solidarity also implies identifying influential Romani women from local communities to adopt the role of facilitators. Simultaneously, shared knowledge between organizations, academic researchers, and RWG will be developed to create trust and facilitate the understanding of cultural norms. Local coalitions will be built, including key stakeholders and policymakers interested in and with expertise on these matters, to support and disseminate RWG’s actions.

#### 2.4.3. Photovoice as an Innovative Way of Doing Things

Romani girls will participate in Photovoice group sessions in order to build critical strengths based on reproductive justice. This process includes identifying and reflecting on their life conditions and those of their communities, interacting with other Romani women role models and building new alliances and support networks, taking photographs regarding their dreams and the resources they need to achieve them, categorizing them to build collective knowledge and strengths, and preparing photo exhibits to advocate for those goals, aspirations, and rights at local, institutional, and policy levels. Romani women facilitators will lead Photovoice implementation along with Romani girls, adopting co-leading roles in decision-making processes and co-assessing its implementation. RJ-based PAR, evaluation, and advocacy capacity will be built among organizations, to ensure the quality, real impact, and sustainability of the actions.

#### 2.4.4. Expected Changes, Outcomes and Impacts

Lastly, regarding impact, changes, and outcomes, qualitative and quantitative indicators will be collected to assess the initiative’s related impact and changes among RWG, Romani grassroots organizations, and their communities.

### 2.5. Participants Timeline

The first four months will be required to map evidence, and to prepare the intervention, developing and adapting guidelines with Romani grassroots organizations, and building local coalitions and collaboration with local communities’ key stakeholders. During the third and fourth month, Romani women facilitators’ recruitment and training will start with a 20-h workshop. This will be followed by the Photovoice process with Romani girls, which will last 6 months. Endline data collection and analysis will ensue in the subsequent 2 months.

### 2.6. Data Collection Methods

Participating girls, significant adults, Romani women facilitators, grassroots organizations, and relevant stakeholders will all contribute to data collection in collaboration with researchers. Romani girls will establish their own goals, processes, and outcomes for all the actions that will be implemented. Afterwards, they will proceed to assess themselves in terms of the implementation of their set goals, action plans, and aspired outcomes. This way, data will be collected, having in mind the intervention steps proposed above (see Table 1; i.e., contextualizing inequities, building partnerships, innovative ways of doing; and expected changes, outcomes, and impacts). External indicators as control mechanisms will be included to guarantee that the decisions that are being made are tailored to local contexts while their quality is ensured.

The RJ-based PAR’s process and implementation evaluation will entail the degree on which planning and logistical activities were considered and adapted by partners to set up and run the intervention effectively [53]. Special focus will be set on assessing how and how many Romani girls and facilitators were identified and selected; how many sessions were organized; which actions were implemented or not, and why; Romani girls’ satisfaction with the process; how they were involved in actions; how they were trained and their permanence was guaranteed. Specific tools will allow us to collect qualitative and quantitative data at individual, organizational, and community levels (see Table 2). To implement Photovoice as an innovative strategy, several Likert scales and open-ended question forms will collect Romani girls’ attendance and significant events within each session. At organizational level, similar forms will collect Romani women facilitators’ perceptions of the Photovoice process while taking stock of each implemented activity in order to maintain the quality of actions through the whole process. To build partnerships, alliances, and solidarity, diverse tools measure social networks (e.g., social convoy at individual level), identify alliances and influential members within the community, and consolidate local coalitions throughout the process (e.g., Local Coalition Ambassador checklist at community level).

Outcome evaluation will assess the results and impact of RJ-based PAR among participants (i.e., Romani girls, facilitators, and Romani grassroots organizations) (see Table 3). In order to contextualize RWG’s inequities and measure Photovoice’s impact, baseline and exit interviews will be carried out among Romani adolescent girl participants and their significant adults (which should be audio-recorded), including qualitative and quantitative data. The comparison group will have the same interviews. Special focus will be placed on evaluating the personal networks of girls, the number of exhibitions organized, and their impacts on the community, and the evolution of Romani girls’ mattering linked to reproductive justice. Concerning organizations, among other aspects, facilitators’ capacity to lead small work groups with Romani girls and conflict-resolution skills will be measured. At a community level, semi-structured interviews will be conducted to map life-stories, community narratives, and community assets. Desk reviews will be developed to identify organizational practices and influence at community and policy levels.

### 2.7. Data Management and Analysis Plan

Research teams will guarantee the anonymity and confidentiality of the information collected. Audio-recordings, transcriptions, and quantitative indicators will be downloaded to password-protected folders with access only by the research partner.

Verbatim transcriptions of interviews and content analysis of qualitative information from both interviews and desk research will be conducted with Atlas-ti 8 software vs. 22 [60] by each research team. Guidelines, followed by a 10-h workshop, will be provided to research partners in order to train them on how to manage quantitative and qualitative data. With regard to data analysis, thematic categories will be generated and analyzed following Corbin and Strauss’ guidelines [61], through an inductive thematic analysis. Thus, a theory will be built derived from data, based on the following steps: (1) data conceptualization to group similar items; (2) naming those groups and defining categories based on their properties and dimensions; and (3) relating those categories to generate the theory. These analyses will be conducted in parallel by two researchers who will analyze the data separately to subsequently reach a consensus, in order to assure inter-observer reliability.

Quantitative data will be analyzed using SPSS vs. 26 [62]. A research partner coordinator will be designated to assure data quality, while each research partner will perform frequencies analysis (of quantitative data) and a 10% random review of input data. Several variables will be controlled, such as date, enrollment at school and academic year, marriage or in a union, pregnancy, number of siblings, position of the girls in relation to their siblings, and family composition. Missing data at item level will be examined using the missing value analysis. A random distribution of the data will be checked according to Little’s MCAR test; if <5% of missing data found per item and <10% of items per scale, the SEM procedure will be performed to impute data. Univariate and multivariate outliers will be examined using box plots and Mahalanobis’ distance, respectively [63]. Equivalence between (1) the intervention group and the comparison group and (2) completers and drop-outs will be examined by performing one-way ANOVAs for quantitative variables and an χ^2^ test for qualitative variables on socio-demographic characteristics and dependent variables at baseline. Statistical assumptions for parametric tests will be checked [64]. Intervention effects will be analyzed at individual level, through the examination of interaction effects from two-way repeated measures ANOVAs (time X group). Differences in the social networks of participant girls will be obtained through qualitative and quantitative changes on Romani girls’ social network (quality of girls’ networks, increases/ decreases in girls’ social network number, reflections on changes attributable to the initiative).

### 2.8. Ethics and Dissemination

Research ethics approval has been obtained from the Andalusian Regional Government (21/2020). The Commission Recommendation of 17 July 2012 on access to and preservation of scientific information, in accordance with the Declaration of Helsinki, will be fulfilled as well as the EU Directive 95/46/EC of the European Parliament and of the Council of 24 October 1995 on the protection of individuals with regard to the processing of personal data and on the free movement of such data.

In accordance with these regulations, an informed consent document will be drawn up for participants to sign, which will reflect these aspects. In the case of pre-adolescents, consent from their parents will also be obtained. This document will include: (a) an explanation of the objectives of the study, its duration, and the time of participation of the subject, who may voluntarily abandon at any time without any negative repercussions; (b) a statement that participation is voluntary and informed; (c) information on the funding of the research, as well as the guarantee that participation will not involve any expense; (d) a description of the benefits for the subjects or third parties; (e) a statement of procedures to ensure data protection, confidentiality, and privacy; (f) the name of the contact person for any questions related to the project; and (g) information on the consequences of the results.

The dissemination plan is an ongoing process that is being implemented at different areas, through the transference of project protocols, results and recommendations within different fields. At academic level, scientific manuscripts will be developed to share obtained results in high impact indexed scientific journals. These manuscripts will be published in open-access journals, in order to guarantee transparency and availability of the generated knowledge for all. The initiative’s conceptual and methodological model proposed, and its results will be also presented at various national and international conferences. At organizational and institutional level, a policy brief on recommendations to adopt a RWG reproductive justice approach will be developed, as well as guidelines, protocols, and a toolbox to be tailored and used in similar contexts, in order to promote the values of reproductive justice in public services and community resources. The methodology proposed in this protocol ensures that conclusions reached throughout the project, as well as developed products, are co-created and co-designed with participants, guaranteeing that the perspectives of the participants are included in the results and that community knowledge transfer and exchange is produced.

## 3. Discussion

In this paper, we present an innovative conceptual and methodological model to empower at-risk Romani girls’ mattering through reproductive justice, aimed at tackling the inequities they suffer. This model describes the process by which PAR inspired by RJ values can create empowering scenarios for RWG and Romani grassroots organizations to learn how to design and implement their own initiatives, as well as learn how to utilize evaluation tools, so they can lead them and decide how to change the course of their own lives while transforming their realities.

Following this model, it is expected that Romani adolescent girls will increase and amplify their personal networks and diversify their sources of social support and influence—including new links with key stakeholders, organizations, and Romani women role models, among others. They will also identify multiple social roles and new possibilities and assets to define their future aspirations; link reproductive justice values with their family, school, and community mattering; increase their socio-political agency to identify community and institutional assets, desire new ones, and advocate for resources; and improve their general satisfaction with life and leadership within decision-making processes. Romani women’s leadership and evaluation capacity will also be fostered through their participation in decision-making processes. In order to achieve these outcomes, meaningful RJ-based PAR must entail developing permanent self-evaluation processes at multiple levels, that allow RWG, communities, and organizations to become self-critical regarding their actions and decisions, and to assess the impact that our activities have on others and in society [43]. Evaluation processes are strongly recognized as powerful and effective strategies to promote the empowerment of the most at-risk populations as they allow the understanding of lived experiences from communities’ own perspectives as accurately and honestly as possible; and then, advocating for improvements at local, regional, and national levels, based on meaningful goals, actions, and credible documentation [65].

At an organizational level, safe spaces that empower RWG’s participation and leadership will be created, promoting Romani women facilitators’ and organizations members’ influence within their local communities, by accompanying Romani girls and their families throughout the initiative’s process to ensure transformative community changes. Accordingly, at community level, alliances and networks will be expanded within local communities, including the consolidation of community ties among girls, their families, and Romani women, and the creation of new networks, including community leaders, key stakeholders, and organizations. For this purpose, Romani organizations must then become scenarios where safe contexts are constructed and promoted, in which RWG ally, expand their networks, and find technical assistance and resources to develop critical knowledge and capacity to advocate for their reproductive rights. This is in line with the recommendations of several institutions, such as the European Parliament, the Agency for Fundamental Rights of the European Union, and the World Health Organization, which include the urge to train organizations and leaders to provide real opportunities for at-risk RWG to lead and define the implementation and evaluation of initiatives in accordance with their priorities [66,67,68].

Lastly, expected changes, outcomes, and impacts at community level also entail the consolidation of Romani organizations’ position as empowering community settings for their members and the local communities they serve [69], as well as their influence at policy and institutional levels towards social change to tackle RWG’s gendered inequities, through the development of organizational advocacy capacity at different levels. Thus, within the proposed model, Romani organizations are posed as perfect scenarios to ensure the active involvement of RWG in self-evaluation processes, and to guarantee the transformation of social contexts and norms [38] and the design and implementation of multilevel advocacy activities [70]. To attain these goals, Romani organizations must become empowering settings for Romani communities, developing the leadership capacity of RWG to ensure that initiatives are tailored to their real needs and interests.

Nonetheless, implementing RJ-inspired PAR is not exempt of potential challenges. First, Romani organizations must ensure that the priorities that are placed at the center of the initiatives’ design and implementation that combat gendered and social inequities are those from local RWG. This implies regenerating organizations to include the voices of the most excluded RWG [52,71]. Furthermore, while evaluation is transformed into a key strategy to facilitate community development and reorganization of power [69], accountability and scientific quality standards must still be guaranteed within all developed initiatives. Rigorous evaluation strategies are needed to assure if or how goals and desired outcomes were met, allowing to demonstrate initiatives’ effectiveness and scientific utility, or to identify areas for improvement [72].

To overcome the aforementioned challenges, organizations are placed as key scenarios to ensure these strategies are meaningful for Romani communities, as well as tailored to real community contexts and conditions [73]. Romani organizations must assure initiatives adhere to quality standards from a rights-centered approach, through the adaptation of initiatives’ implementation and evaluation as strategies grounded on the voice and real interests of RWG [74]. Adopting this perspective ensures co-creating evidence-based initiatives that are placed at the service of Romani communities and organizations, and at the same time, they can become processes that guarantee transformative social changes [70]. This proposal is expected to ensure that most at-risk communities have spaces to meaningfully advocate for the full enjoyment of their rights from effective, significant, and sustainable proposals that have a real impact on their lives.

## 4. Conclusions

In summary, this initiative seeks to contribute to overcoming the inequities suffered by RWG associated with their reproductive decisions by focusing on three key points:

1. To transcend the dominant rhetoric that—associating reproductive health to “health literacy”, “family planning”, “contraception”, “pregnancy termination”—becomes the perfect ally of slow violence induced by systemic anti-Gypsyism;

2. To expose the political intentionality of the ostracism suffered by the Romani population through the instrumentalization of the female reproductive capacity—a key vector for the social reproduction of health inequalities. This initiative emphasizes how reproduction is a central axis of the multiple and intersectional oppressions suffered by the Romani population;

3. To consider Romani girls as political subjects who make informed decisions about their lives, find in the women of their communities the natural allies to achieve them, and expand their networks according to their interests and goals. This proposal seeks to facilitate spaces for Romani women to transform their gender narratives into a vector of alliance between different generations to achieve their aspirations of a prosperous life for themselves, their loved ones, and their communities.

## Figures and Tables

**Table 1 healthcare-11-00755-t001:** Reproductive justice participatory action research for Romani girls’ mattering.

	Individual	Organizational	Community
Contextualizing Romani women and girls’ inequities	Girls’ life satisfaction, sense of mattering, and agency linked to gender roles. Social convoy based on family ties	Mapping organizational settings and narratives, practices, and community influence	Life narratives to identity experiences of oppression, resistance, and resilience
Building partnerships, alliances, and solidarity	Building complicity and alliances between girls, peers, and significant adults	Building local coalitions and collaborative capacity with key stakeholders at multiple levels	Identifying community assets and influential members within the community
Photovoice as an innovative way of doing things	Photovoice to envision aspirations and resources to achieve them, planning activities to advocate for their rights	Ensuring RWG leadership within organizations, facilitating Photovoice and advocacy activities, and adapting evaluation processes	Maximizing the dissemination and impact of advocacy processes at community, institutional, and policy levels
Expected changes, outcomes, and impacts	Girls and women’s reproductive justice-based mattering, agency, and life satisfaction. Enlarged social convoy	Romani women facilitators’ increased recognition in the community, and leadership to design actions, implement, and evaluate initiatives	Romani associations as empowering community settings for RWG, radiating influence and leading transformative policy changes based on Romani strengths

**Table 2 healthcare-11-00755-t002:** Process and implementation evaluation indicators.

	Tool	General Characteristics	Content
Individual level	Wrap-up notes	Open-ended questions	Significant event(s) and satisfaction of Romani Girls in the Photovoice sessions
	Romani girls’ attendance sheets (ad hoc)	Quantitative scales to assess participation	Through 4-point Likert scales, overall participation/adherence level
Organizational level	Local Coalitions’ attendance (ad hoc)	Quantitative scales to assess attendance of activities	Through 4-point Likert scales overall participation level for each member attending each activity individually
	Quality Assurance Checklists on evaluation capacity (ad hoc)	Self-assessment tool to measure Romani women facilitators’ decision-making, leadership, and evaluation capacity	Eleven open questions for Romani women facilitators to reflect on their work to help them build their skills and continuously improve in leading the sessions, analyzing the skills and knowledge they have acquired, and areas where they might need additional support
	Romani women facilitators’ meeting notes	Quantitative scales to assess sessions’ emotional climate, and collect girls’ reactions	Date, agenda points, discussed topics, and decisions made, as well as overall emotional climate of the session through 4-point Likert scale
	Facilitation Skills (ad hoc)	Self-assessment questions to collect facilitators’ reflections	A 10-point Likert scale through 3 items on facilitation skills and 5 open ended questions about areas where they might need more support
	Action Plan Forms (ad hoc)	Open-ended questions that will help the facilitators and Romani girls to map out and plan	Eight open-ended questions aimed at designing the actions, participants, setting the date, location, and goals for the activity/initiative and outlining each step that needs to be taken, anticipate barriers and strategies to mitigate them
Community level	Local Coalition Ambassador checklist (ad hoc)	Open-ended questions to identify relevant policymakers, leaders, and service providers	Date of meeting/conversation, name(s) of person(s), organization, position of the person, involvement in the project, and follow up actions to disseminate and build networks to support Romani girls’ activities
	Advocacy Plan and attendance sheets (ad hoc)	Open-ended questions to collect attendance, and decisions made during activities	Number of exhibitions and other advocacy activities developed, number and position/role of people reached and involved within the activities

**Table 3 healthcare-11-00755-t003:** Outcome evaluation indicators.

	Tool	General Characteristics	Content
Individual level	Social Convoy [54]	Semi-structured interviews to assess personal networks	Includes three levels—i.e., inner circle (closest people), middle circle (people they care about, but not so close) and outer circle (weak ties)
	Societal Mattering Scale [55]	A 4-point Likert scale to measure girls’ sense of mattering	Eighteen items to measure mattering in their community and school (e.g., “The people in my community value me as a person”) and nine items to measure mattering within family contexts (e.g., “The people in my family value me as a person”)
	Life Satisfaction Scale [56]	A 7-point Likert scale to measure satisfaction with life	Five items (e.g., “I am satisfied with my life”)
	Policy Control Subscale [57]	Quantitative scale to measure Romani girls’ policy control	Four items (e.g., “Youth like me have the ability to participate effectively in community activities and decision making”)
	Relevance of Identity Roles Scale (ad hoc)	A 4-point Likert scale to measure relevant roles for Romani Girls	Eight items (e.g., “How important to you in obtaining your life goals is finishing high school”)
	Baseline and endline interviews with Romani girls and their families	Semi-structured interviews (ad hoc)	Twenty open-ended questions regarding family context, interests, analytical skills and leadership abilities (e.g., What types of games do you like to play with your friends? Who is the one who decides what to play?).
Organizational level	Baseline and endline interviews with Romani facilitators	Semi-structured interviews (ad hoc)	Twenty-five open-ended questions re. information on Romani women facilitators’ community work experience with minors
	Baseline and endline interviews with members of the organization	Semi-structured interviews (ad hoc)	Organization members’ experience, and motivation towards organizations’ goals. Organizational opportunities for members to participate, core activities or adaptation to local contexts
	Desk reviews	Collection of materials to map community influence and assets	Good practices and organizations’ recognition and influence at community, institutional, and policy levels
Community level	Map life-stories and community narratives [39,58]	Stakeholders and recipients’ semi-structured interviews	Organizations radiating influence to local communities as well as stakeholders and policymakers’ perceptions of organizations’ social impact at local and institutional levels
	Map community assets [59]	Organization and stakeholders’ semi-structured interviews	Numbers of services and resources used at local communities, type of services used, satisfaction with each service, frequency of use, access barriers, and other desired resources

## Data Availability

No new data were created or analyzed in this study. Data sharing is not applicable to this article.

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
