# Peer review of "Romani Girls Matter: Developing a Participatory Action Research Protocol for Reproductive Justice"

_healthcare, 2023, doi:10.3390/healthcare11050755_

Round 1

Reviewer 1 Report

The paper is exellent in design and clarity, the topic crucial to let social justice be concrete.

The research design must be better described to explain how the participants have been involved in the very initial phase and problem setting, before start to act for problem solving. I suggest to take a look at Action research juornal to go more in deep in the methodological aspects and find usual literature support.

Then, I suggest to take into consideration the concept of slow violence, for instance as it is proposed by Nixon, R. (2011). Slow Violence and the Environmentalism of the Poor. Harvard University Press or Horsti, K., & Pirkkalainen, P. (2020). The slow violence of deportability. In Violence, gender and affect: Interpersonal, institutional and ideological practices (pp. 181-200). Cham: Springer International Publishing to offer a deeper understanding of the risks correctly pointed in lines 443-444. I think that this issue must be better problematized.

Please consider the comments in the attached file.

Author Response

Dear reviewer,

Thank you for these suggestions. We have carefully considered the comments, which have contributed to improve the manuscript.

Please, see the attachment in which we describe all the modifications of the updated version of the paper.

Reviewer 2 Report

Review of submission to Healthcare: ‘Reproductive justice for Romani Girls’ Mattering: A participatory action research protocol’.

February 2023

This article is neither a research article nor a review article. It is not a case report. It rather comprises a research proposal, or ‘protocol’.

As I was reading it, I started to wonder what I am actually reviewing – am I considering the merits of a description of a research process that involved the development of a research protocol or am I considering an actual research proposal. If the former, it needs to be re-oriented to reflect this, if the latter it seems inappropriate for a journal article.

My main impression was that I’m reading the latter, a research proposal that, for example, might be submitted to a granting organisation for funding.

I then wondered if this material is of interest to the relevant research community, and concluded that yes, there is certainly a case for inclusion in a research journal.

Therefore, my assessment is that the presentation of the content needs to be ‘re-oriented’ to clarify a justifiable rationale, to enable the status of a journal article.

To achieve this, I suggest it be introduced with a clear exposition of the aim of the paper, for example, to describe the result of a process undertaken for the purpose of xyz… The authors give evidence that the process was funded by a grant(s). So this means outlining what the call for bids was requiring or seeking, what this call for proposals was aiming to achieve, etc; then a brief outline of the actual proposal-to-develop-a-protocol that was submitted by the authors; then a statement to the effect that their proposal was accepted and funded; and then how the authors’ research protocol was responding to the aims of the original call for proposals.

The main body of the article would be the presentation of the actual protocol, as currently described. The conclusion also needs to be adapted accordingly.

This would give a context for the reader, that is currently missing, and would then create a justifiable contribution as a journal article. It might be that the process itself was different from that described here (I’m just guessing), but whatever happened needs to be described.

The actual research protocol is itself impressive. If I was assessing this for a relevant grant funding round, I would rank it highly. It has a clear political intent that is well-stated, justified, consistently pursued, and of significant importance. The goals are realistic while at the same time pushing boundaries. The design, methods and materials are appropriate, robust, comprehensive, and carefully and thoroughly worked through to arrive at a solid foundation for a highly worthwhile project. This could become a model for further extrapolation of such work, if the project itself is funded and succeeds. It should be funded and has every chance of success, especially given the commitment of the researchers to an uncompromising focus on an empowering strategy that is not just paying ‘lip-service’ to this idea but actually grounds the entire research process in such an ethos.

The standard of English language is high, with very few problems evident. The title, however, does seem a little clumsy, and potentially ambiguous, to me. The expression ‘Girls’ mattering’ creates something of a stumbling block, and I’d suggest a slight change, for example: ‘Romani Girls Matter: Developing a Participatory Action Research Protocol for Reproductive Justice’.

There are a few minor points to attend to:

Line 39: ‘obstacles to access to a regular labor market’ would be better as ‘obstacles to accessing a regular labor market’

Line 61: ‘causes of their own suffering, and perceive themselves as entitled…’ better as ‘causes of their own suffering, perceiving themselves as entitled…’

Line 64: I don’t think ‘tabaquism’ is an English word – can the appropriate English term be substituted, or meaning footnoted?

Line 83: ‘feminist theory’ doesn’t sit well in this list. I think, given the nature of the other items, this could be better worded as ‘feminist struggle against sexism’, or ‘feminist issues’, or ‘gender equity’ etc.

In summary, the development of this protocol offers a valuable contribution to the literature on participatory action research for community development, community psychology, women’s health, Romani health justice, and regarding concerns with equity and participation in major public health issues in the healthcare context. Re-oriented as suggested, publication would bring this work to the attention of other researchers who, I’m sure, will then not only find it useful, but would also be interested to follow the further implementation of this project and its progress.

The statement of ‘Conflicts of Interest’ appended at the end of the submission implies that this research protocol has actually been implemented, data interpreted etc. If this is the case, this needs to be clarified, because no ‘interpretation of data’ appears here. Given the content as presented, the ‘conflicts of interest’ statement needs to align with the content of the (re-oriented) paper, to avoid any confusion.

Author Response

(The authors gave the same response as above.)

Round 2

Reviewer 1 Report

Congratulations for the good work done in upgrading the article. Now it seems to be ready for publication.